# Electrochemical Detection of Ascorbic Acid in Oranges at MWCNT-AONP Nanocomposite Fabricated Electrode

**DOI:** 10.3390/nano12040645

**Published:** 2022-02-15

**Authors:** Pholoso Calvin Motsaathebe, Omolola Ester Fayemi

**Affiliations:** 1Department of Chemistry, Faculty of Natural and Agricultural Sciences, Mafikeng Campus, North-West University, Mmabatho 2735, South Africa; pholosocalven@gmail.com; 2Material Science Innovation and Modelling (MaSIM) Research Focus Area, Faculty of Natural and Agricultural Sciences, Mafikeng Campus, North-West University, Mmabatho 2735, South Africa

**Keywords:** ascorbic acid, oranges, electrochemical sensors, nanocomposite, antimony oxide nanoparticles

## Abstract

Ascorbic acid (AA) is an essential vitamin in the body, influencing collagen formation, as well as norepinephrine, folic acids, tryptophan, tyrosine, lysine, and neuronal hormone metabolism. This work reports on electrochemical detection of ascorbic acid (AA) in oranges using screen-print carbon electrodes (SPCEs) fabricated with multi-walled carbon nanotube- antimony oxide nanoparticle (MWCNT-AONP) nanocomposite. The nanocomposite-modified electrode displayed enhanced electron transfer and a better electrocatalytic reaction towards AA compared to other fabricated electrodes. The current response at the nanocomposite-modified electrode was four times bigger than the bare electrode. The sensitivity and limit of detection (LOD) at the nanocomposite modified electrode was 0.3663 [AA]/µM and 140 nM, respectively, with linearity from 0.16–0.640 μM and regression value R^2^ = 0.985, using square wave voltammetry (SWV) for AA detection. Two well-separated oxidation peaks were observed in a mixed system containing AA and serotonin (5-HT); and the sensitivity and LOD were 0.0224 [AA]/µA, and 5.85 µΜ, respectively, with a concentration range from 23 to 100 µM (R^2^ = 0.9969) for AA detection. The proposed sensor outperformed other AA sensors reported in the literature. The fabricated electrode showed great applicability with excellent recoveries ranging from 99 to 107 %, with a mean relative standard deviation (RSD) value of 3.52 % (*n* = 3) towards detecting AA in fresh oranges.

## 1. Introduction

L-Ascorbic acid (AA) or ascorbate, commonly known as vitamin C, is an important water-soluble vitamin derived from green vegetables, fruits, and other dietary supplements [1]. The role played by AA in the immune defence is extensive [2]. It enables collagen synthesis, which is needed to maintain healthy bones, teeth, skins, cartilages, and stabilization [3]; reduces histamine levels [4]; regulates cytokine synthesis [5]; enhances antibody levels and acts as an antioxidant [6]; reduces necrosis and facilitates apoptosis [7]; improves phagocytosis [8]; and speeds up the wound healing process [9]. In addition, the function of AA as an enzyme co-factor has been established [10,11]. 

Abnormal AA levels in bodily fluids have been reported to cause cancer, cardiovascular diseases, and Alzheimer’s and Parkinson’s diseases [12,13]. Extended use of AA could cause urinary oxalate calculus, increase infertility in a woman, and affect embryo development [14]. Furthermore, excessive AA use has been reported to cause diarrhoea, nausea, vomiting, headache, insomnia, gastric irritation, renal problems, loss of food taste, and vomiting. Moreover, excess amounts of AA can act as a peroxidation in the presence of heavy metals such as iron [15]. 

Although important for their wellbeing, humans, guinea pigs, and other species of birds, fishes, and mammals cannot produce AA. The inability of these aforementioned species to produce AA is attributed to their evolution over time and specifically the mutation of the L-gulono-lactone oxidase gene, which codes for the enzyme responsible for speeding up the final phase of vitamin C synthesis [16,17]. People obtain their AA supplement from fresh fruits, such as oranges, lemons, mangoes, papayas, strawberries, blackberries, and guava, and vegetables, such as tomatoes, carrots, potatoes, cabbage, and spinach. However, alcohol abuse, drug abuse, ageing, the development of diseases, such as anorexia, and poor diet are among the leading reported causes of reduced AA levels in the body [17,18]. 

Due to their importance to the food, cosmetic, chemical, and pharmaceutical industries, various analytical methods, such as coulometric [19], high performance liquid chromatography (HPLC) with electrochemical colourimetry [20], electrochemical [21], amperometric [22], capillary electrophoresis [23], gas chromatography [24], and liquid chromatography [25], have been used to determine AA. Nonetheless, most of these analytical methods were found to be less sensitive, expensive, complicated processes and require trained personnel to use them. They require longer sample preparation, are time-consuming; moreover, others need laboratory equipment, except for electrochemical technique [26,27]. Electrochemical sensors fabricated with carbon nanotubes (CNTs), metal oxide nanoparticles, and graphene oxide and conducting polymers have been used extensively to detect AA in different real-life samples [28,29,30,31,32,33,34,35]. These sensors are found to be affordable, offer quick and efficient analysis, and offers highly sensitive and selective.

Nanomaterials have played a critical role in developing affordable, sensitive, and selective sensors [36]. Antimony oxide nanoparticles (AONPs) have found a wide application in the chemical, semi-conducting, and sensing industries due to their unique and exceptional properties, including a high refractive index, high abrasion resistance, high proton conductivity, superior mechanical strength, and high photon absorption capacity [37,38]. They have been used to make flame retardant synergists [39,40], humidity sensors [41], gas sensors [42], light-emitting diodes (LEDs) [43], binary glasses [44], solar cells [45], and anti-friction alloys [46].

The widespread adoption of MWCNTs as electrode modifiers is based on their outstanding and remarkable properties, which include high electrical conductivity, unique mechanical and structural properties [36,47]. CNTs are acid-functionalized to enhance electron transfer and assist with adsorption [48]. CNTs reinforced with metal oxide nanoparticles nanocomposite fabricated electrodes are increasingly being investigated due to improved electrical, magnetic, and optical properties resulting from synergies between two nanomaterials [49,50].

Apinya et al. [51] developed a polyaniline/MnO_2_-Sb_2_O_3_ nanocomposite sensor to detect AA and acetylsalicylic acid (ASA) in human urine. Using the differential pulse voltammetry (DPV) technique, their sensors demonstrated sensitivity and LOD values of 0.9356 nmol L^−^^1^ and 0.071 nmol L^−^^1^, respectively, with a concentration range within 0.1–372.64 nmol L^−^^1^ (R^2^ = 0.9974). In the same vein, Şehriban et al. [52] used the SWV technique to determine metaproterenol from biological and pharmaceutical materials using AONP-WO_3_-CNTs@GCE. Their electrode displayed a linear range and LOD of 52–270 nM for metaproterenol detection and 6.6 nM. Similarly, Masibi et al. [53,54] used MWCNT-AONP-PANI@GCE to detect endosulfan, and lindane, respectively, from water samples via the SWV method.

No research paper in the literature discusses the determination of AA in tomatoes using AONP-MWCNT nanocomposite electrodes. Therefore, this paper discusses the detection of AA at nanocomposite by employing cyclic voltammetry and square wave techniques. The fabricated sensors displayed enhanced electrocatalytic and electroanalytic properties towards detecting AA. The synergistic effect of the two nanomaterials resulted in faster electron transfer kinetics at the electrode.

## 2. Reagents and Experimental Procedure

### 2.1. Reagents

The following chemicals were purchased and used as received from the manufacturers, serotonin (5-HT) hydrochloride powder (98%), antimony trichloride (99%) (SbCl_3_), ascorbic acid (AA) (99%), hydrochloric acid (32%) (HCL), sodium hydroxide (NaOH) (98%), multi-walled carbon nanotubes abbreviated MWCNTs (≥98%), N, N-dimethylformamide (C_3_H_7_NO) (99%), nitric acid (HNO_3_) (34%), and toluene (C_7_H_8_) (99%) were obtained from Merck Pty Ltd. (Darmstadt, Germany). Sodium phosphate salts Na_2_HPO_4_ (99%) and NaH_2_PO_4_ (99%) products of LABCHEM and GlassWorld, located in Johannesburg, South africa were used in the preparation of 0.1 M phosphate buffer solution (PBS) at pH 7. Emplura^®^ Merck (The Chemical Center from Mumbai, India) provided distilled water, which was used through the experiment to prepare chemicals. 

### 2.2. Antimony Oxide Nanoparticles (AONPs) Synthetic Method

The synthetic method followed to synthesis antimony oxide nanoparticles (AONPs) was adopted from Chen et al. [55]. A colourless solution was prepared by dissolving 2 mM antimony trichloride in toluene under rapid stirring for 15 min. Then, 20 mL of distilled water was added to create a lacteous colloid. Then the solution was re-adjusted to a pH of 8–9 by adding 6 M NaOH. It was then agitated for an additional 20 min and then placed in a stainless-steel autoclave with Teflon-lined walls for 12 h at 120 °C. The formed product was washed with distilled water and ethanol several times and then dried in an oven at 60 °C for 6 h.

### 2.3. Synthesis of f-MWCNTs

The nitric acid treatment approach synthesized functionalized multi-walled carbon nanotubes (f-MWCNTs). In 1 M nitric acid, about 300 mg of raw multi-walled carbon nanotubes (r-MWCNTs) was dissolved. After that, the mixture was sonicated in cold water at 50 °C for 240 min. The mixture was then filtered with deionized water until a pH 7 was reached. The finished product was dried in an oven overnight [56].

### 2.4. MWCNT-AONP Nanocomposite Synthesis

About 2 mg f-MWCNTs and 6 mg AONPs were dissolved into 2 mL dimethylformamide (DMF) with continuous stirring at room temperature for 48 h. The resultant AONP-MWCNT nanocomposite was dried at 25 °C in an oven for 24 h [55].

### 2.5. Real-Sample Analysis

The applicability of the fabricated electrode to detect AA in a real sample was examined using fresh oranges bought from a nearby supermarket. The oranges were crushed to extract their juice and filtered using Whatman No 1 filter paper to obtain an AA extract solution. A fixed volume of the oranges extract (1 mL) was added to different volumes of the stock solution to make up to 10 mL of the sample. The experiment was done in triplicates using SWV.

## 3. Results and Discussion

### 3.1. Electrocatalytic Experiements

The comparative cyclic voltammogram between bare and modified screen printed carbon electrodes (SPCEs) in 0.1 mM AA is shown in Figure 1. The voltammograms in Figure 1a indicate that the reactions of AA at the electrodes were irreversible; also, modified electrodes showed enhanced cyclic voltammogram shapes compared to bare electrodes. The voltammogram in Figure 1a,b showed a decrease in AA current response in the following order: SPCE-AONP-MWCNT (Ipa = 60.71 µA; Epa = 0.032 V), SPCE-AONPs (Ipa = 11.36 µA; Epa = 0.034 V), and SPCE-bare (Ipa = 14.96 µA; Epa = 0.45 V). The Epa value of 0.032 V at the nanocomposite modified electrode matched with other studies from the literature [28,29,57], while the SPCE/f-MWCNTs did not interact with AA. A higher current response noted at the nanocomposite electrode was attributed to the synergic effect between the two nanomaterials. Figure 1c shows a scan rate study at the proposed sensor with a scan rate of 10–350 mVs^−^^1^. According to Figure 1d, a diffusion-controlled reaction was based on a linear relationship against peak current vs. square root. The linear relationship between Ipa vs *v*^1/2^ can be expressed with this equation: Ipa = 44.143 *v*^1/2^ − 135.55 (R^2^ = 0.9930). A detailed description of the electrooxidation mechanism pathway of AA is provided by Figure 1.

### 3.2. Electro-Analytic Experiments

Figure 2a,b depicts a linear relationship between peak current and increasing AA (0.040–0.640 µM) concentration. For AA detection, the sensitivity, LOD, and limit of quantification (LOQ) was 0.3663 [AA]/µA, 140 nM, and 423 nM, respectively, with concentration range from 4 × 10^−^^8^–6.4 × 10^−^^7^ M (R^2^ = 0.9851). As shown in Table 1, the proposed sensor performed better than other AA electrochemical sensors from the literature. The synergy between MWCNTs and AONPs in facilitating electron transfer cannot be overstated. Similarly, the MWCNTs are an essential component of the nanocomposite electrode’s outstanding performance. Its wide surface area allows free passage of electrolytes and charges between the base electrode and electroactive species on the electrode surface. The conductive properties of the AONPs and MWCNTs and the ionic interaction between the two nanomaterials are some of the mechanisms responsible for this considerable electron transport at the fabricated electrode. Using Equations (1) and (2), the suggested electrode’s resulting LOD, and LOQ values were computed. The results suggest that the constructed sensors detect AA at low concentrations with excellent sensitivity and reliability.
(1)$$\mathrm{LOD}=\frac{3.3\times SD}{s},$$
(2)$$\mathrm{LOQ}=\frac{10\times SD}{s},$$

*SD* stands for standard deviation, while *s* stands for gradient slope.

### 3.3. Simultaneous Detection of AA and 5-HT

Figure 3a shows simultaneous detection of AA (23–100 µM) in the presence of 0.5 mM 5-HT. The AA current response increased linearly with increasing AA concentration while the 5-HT peak current remained constant. It was evident that the proposed sensor has excellent anti-interference performance, as evidenced by two visible well-separated peaks located at 0.038 V and 0.350 V representing AA and 5-HT, respectively, as well as a 360 mV potential peak separation between AA- 5-HT as displayed in Figure 3c. Figure 3b shows corresponding calibration plots for AA. For detection of AA in presences of 0.5 mM 5-HT, the sensitivity, LOD, and LOQ at the modified electrode were 0.0224 [AA]/µA, 5.85 µΜ, and 19.51 μM, respectively, with a linear ranging from 23–100 µM (R^2^ = 0.9969) utilizing SWV. 

### 3.4. Reproducibility and Shelve-Life Study of The Proposed Sensor

This experiment aimed to test the feasibility and re-usability of the constructed sensor. Using the CV technique, the experiment was repeated (20 times) in 0.1 mM AA at a scan rate of 25 mVs^−^^1^. The anodic peak decreased by 42%, and the electrode showed an RSD value of 20% towards the detection of AA, as shown in Figure 4a. A decrease in the AA peak current might be attributed to more nanocomposite (SPCE-MWCNT-AONP) assimilation into the electrode surface over time. Figure 4b shows the results of the repeatability study at the suggested electrode for four weeks using the cyclic voltammetry (CV) method. The constructed sensor was stored in a dry environment when not in use. The electrode maintained an acceptable AA current response for 4 weeks, indicating the proposed sensor’s longevity. 

### 3.5. Real-Aample Analysis 

The proposed electrode displayed excellent recoveries ranging from 99.12 to 107.76%, with an average RSD (%) value of 3.52 (*n* = 3) for detecting AA in oranges. The results of the experiment are summarised in Table 2. The % recovery was calculated utilizing Equation (3).
(3)$$\%\;\mathrm{Recovery}=\frac{\mathrm{quantity}\;\mathrm{found}\;–\;\mathrm{quantity}\;\mathrm{detected}\;}{\mathrm{quantity}\;\mathrm{added}\;}$$

## 4. Conclusions

This study has finalized the development of an electrochemical sensor fabricated with AONP-MWCNT on SPCEs to detect AA in tomatoes. The nanocomposite electrode displayed greater electrocatalytic activity toward AA with respect to other electrodes. The reactions of AA are diffusion-controlled and irreversible at the nanocomposite electrode. For AA detection, the proposed sensor electrode displayed sensitivity, LOD, and LOQ values of 0.4789 µA/µM, 278 nM, and 842 nM, respectively, with linearity from 0.16 to 0.640 μM (R^2^ = 0.9851) using SWV. For simultaneous detection of AA in the presences of 0.5 mM 5-HT, the electrode displayed the sensitivity, LOD, and LOQ as 0.0224 µA/µM, 5.85 μM, 19.51 μM, respectively, with linearity from 23 to 100 μM (R^2^ = 0.9969). The fabricated sensor exhibited excellent anti-interference behaviour and long shelve life. A sensor based on the proposed design was successfully applied to detecting AA in oranges, showing remarkable recovery rates exceeding 99.12% with a 3.52% relative standard deviation (RSD). The proposed sensor is highly selective, simple to use, inexpensive, quick, and offers accurate analysis. Therefore, this sensor could potentially be used to detect other important analytes such as neurotransmitters, organochlorides (pesticides and herbicides), and vitamins, etc. For future studies, a binder or electrode stabilizers could be used to improve the stability of the proposed sensor.

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
