# Peer review of "Electrochemical Detection of Ascorbic Acid in Oranges at MWCNT-AONP Nanocomposite Fabricated Electrode"

_nanomaterials, 2022, doi:10.3390/nano12040645_

Round 1

Reviewer 1 Report

In this work, electrochemical detection of AA in oranges is performed using screen-print carbon electrodes fabricated with MWCNT-AONP nanocomposite, and the fabricated electrode showed great applicability with excellent recoveries. These are important findings; however, the paper requires major revision before it can be considered for publication, pending satisfactory addressing of the following comments:

  1. In abstract, it is necessary to supplement a brief introduction of AA for better understanding. For example, "INTERNATIONAL JOURNAL OF ELECTROCHEMICAL SCIENCE, 2020, 15 (4), 3327-3346".
  1. Source of all chemicals used should be specified.
  1. In table 1, the proposed sensor performed better than other AA electrochemical sensors from the literature. Please clarify that why the sensor fabricated with MWCNT-AONP nanocomposite performed better than other sensors on mechanism.
  1. In line 207, the figure is named as Figure 3, which is the same as the figure in line 184. These two figures are different, and their titles should be distinguished. In Figure 3(a) in line 204, the values of AA concentration of should be provided.
  1. The format of Figure 2 should be further improved referring to Figure 1.
  1. The form and style of the conclusion need improvement. Conclusion serves as some kind of long resumé of the efforts, the significance and inspiration of this work, and provides perspective of future routes in this field, instead of only serving the reader with a few highlights of the study which is usually less important.

Author Response

The Editor

Nanomaterials

REVISION OF MANUSCRIPT SUBMITTED FOR PUBLICATION

Manuscript Title:  Electrochemical detection of ascorbic acid in oranges at MWCNT-AONP nanocomposite fabricated electrode.

Manuscript ID: nanomaterials-1580029

We appreciate the reviewers' comments on our paper and have submitted the updated version for further consideration. Indeed, the remarks enhanced the quality of our paper.

We have addressed each point raised by the reviewers. All reviewers' concerns and ideas have been thoroughly considered and implemented. The authors' responses to the reviewers' comments and suggestions are included below. The reviewers' comments and recommendations are written first, followed by the writers' reply, which are written under the heading "Authors' response". Also, please note that the updated manuscript's altered sections are indicated in yellow.

REVIEWER 1

Comment 1: In abstract, it is necessary to supplement a brief introduction of AA for better understanding. For example, "INTERNATIONAL JOURNAL OF ELECTROCHEMICAL SCIENCE, 2020, 15 (4), 3327-3346".

Authors’ response: Thank you for the comment A brief introduction of AA to enhance understanding has been added to the session using the example given above. See section highlighted in yellow under the abstract section.

 Comment 2: Source of all chemicals used should be specified.

Authors’ response: All the source of the chemicals have been noted.

Comment 3: In table 1, the proposed sensor performed better than other AA electrochemical sensors from the literature. Please clarify that why the sensor fabricated with MWCNT-AONP nanocomposite performed better than other sensors on mechanism.

Authors’ response: Thank you for this input. A justification have been presented to why the MWCNT-AONP nanocomposite fabricated sensor performed better than other electrochemical sensors from literature based on the mechanism of the sensor. See highlighted section under 3.2. Electro-catalytic experiments section.

Comment 4: In line 207, the figure is named as Figure 3, which is the same as the figure in line 184. These two figures are different, and their titles should be distinguished. In Figure 3(a) in line 204, the values of AA concentration of should be provided.

Authors’ response: Thank you for your kin observation. The error has been corrected and the numbering corrected and the titles have been distinguished. In Figure 3(a) in line 204, the values of the AA concentration have been added. See Figure 3(a).

Comment 5: The format of Figure 2 should be further improved referring to Figure 1.

Authors’ response: Thank you for such valuable input. The format of Figure 2 has been improved.

Comment 6: The form and style of the conclusion need improvement. The conclusion serves as some kind of long resumé of the efforts, the significance and inspiration of this work, and provides perspective of future routes in this field, instead of only serving the reader with a few highlights of the study which is usually less important.

Authors’ response: Thanks. The form and style of the conclusion have been revised and improved as advised. See the highlighted section in the conclusion section.

Reviewer 2 Report

In this paper, MWCNT-AONPs nanocomposites were prepared by a hydrothermal reaction method, and the materials were used as electrodes for sensing detection of ascorbic acid AA. The sensing performance of MWCNT-AONPs was tested repeatedly and the maximum lifetime was analyzed by methods such as cyclic voltammetry and square wave technology. However, the structure and morphology of AONPs have not been characterized. The corresponding data, such as SEM, TEM, need to be supplemented, and it is necessary to analyze whether the morphology and structure of the nanomaterials have changed. That is, the stability should be studied. And, the performance comparison with other nanomaterials should be tabled. Other works about the detection of pollutants using electrochemical methods could be read and discussed, such as Dalton Trans., 2021,50, 15567.

Author Response

The Editor

Nanomaterials

REVISION OF MANUSCRIPT SUBMITTED FOR PUBLICATION

Manuscript Title:  Electrochemical detection of ascorbic acid in oranges at MWCNT-AONP nanocomposite fabricated electrode.

Manuscript ID: nanomaterials-1580029

We appreciate the reviewers' comments on our paper and have submitted the updated version for further consideration. Indeed, the remarks enhanced the quality of our paper.

We have addressed each point raised by the reviewers. All reviewers' concerns and ideas have been thoroughly considered and implemented. The authors' responses to the reviewers' comments and suggestions are included below. The reviewers' comments and recommendations are written first, followed by the writers' reply, which are written under the heading "Authors' response". Also, please note that the updated manuscript's altered sections are indicated in yellow.

REVIEWER 2

In this paper, MWCNT-AONPs nanocomposites were prepared by a hydrothermal reaction method, and the materials were used as electrodes for sensing detection of ascorbic acid AA. The sensing performance of MWCNT-AONPs was tested repeatedly and the maximum lifetime was analyzed by methods such as cyclic voltammetry and square wave technology. However, the structure and morphology of AONPs have not been characterized. The corresponding data, such as SEM, TEM, need to be supplemented, and it is necessary to analyze whether the morphology and structure of the nanomaterials have changed. That is, the stability should be studied. And, the performance comparison with other nanomaterials should be tabled. Other works about the detection of pollutants using electrochemical methods could be read and discussed, such as Dalton Trans., 2021,50, 15567.

Authors’ response: Thank you for your insightful input and contribution to our work. The corresponding data, such as SEM, TEM have been reported and discussed in our previous study based on the use of the catalyst for serotonin detection in tomatoes.

Round 2

Reviewer 1 Report

The revised manuscript can be accepted for publication.